# Feeling the Heat: The *Campylobacter jejuni* HrcA Transcriptional Repressor Is an Intrinsic Protein Thermosensor

**DOI:** 10.3390/biom11101413

**Published:** 2021-09-27

**Authors:** Giovanni Versace, Marta Palombo, Anna Menon, Vincenzo Scarlato, Davide Roncarati

**Affiliations:** Department of Pharmacy and Biotechnology (FaBiT), University of Bologna, 40126 Bologna, Italy; giovanni.versace@studio.unibo.it (G.V.); marta.palombo2@unibo.it (M.P.); anna.menon5@studio.unibo.it (A.M.)

**Keywords:** transcriptional regulation, DNA-protein interaction, heat-shock response, HrcA repressor, HspR repressor, GroE chaperonin, heat-shock proteins, signal perception, *Campylobacter jejuni*

## Abstract

The heat-shock response, a universal protective mechanism consisting of a transcriptional reprogramming of the cellular transcriptome, results in the accumulation of proteins which counteract the deleterious effects of heat-stress on cellular polypeptides. To quickly respond to thermal stress and trigger the heat-shock response, bacteria rely on different mechanisms to detect temperature variations, which can involve nearly all classes of biological molecules. In *Campylobacter jejuni* the response to heat-shock is transcriptionally controlled by a regulatory circuit involving two repressors, HspR and HrcA. In the present work we show that the heat-shock repressor HrcA acts as an intrinsic protein thermometer. We report that a temperature upshift up to 42 °C negatively affects HrcA DNA-binding activity to a target promoter, a condition required for de-repression of regulated genes. Furthermore, we show that this impairment of HrcA binding at 42 °C is irreversible in vitro, as DNA-binding was still not restored by reversing the incubation temperature to 37 °C. On the other hand, we demonstrate that the DNA-binding activity of HspR, which controls, in combination with HrcA, the transcription of chaperones’ genes, is unaffected by heat-stress up to 45 °C, portraying this master repressor as a rather stable protein. Additionally, we show that HrcA binding activity is enhanced by the chaperonin GroE, upon direct protein–protein interaction. In conclusion, the results presented in this work establish HrcA as a novel example of intrinsic heat-sensing transcriptional regulator, whose DNA-binding activity is positively modulated by the GroE chaperonin.

## 1. Introduction

All living organisms are continuously challenged by several stress factors, which represent an obstacle to life and threaten their survival. Among the different environmental insults, a temperature increase only moderately above the organism optimum growth condition poses a major threat, leading to cell damage or death. Indeed, a sudden increase of environmental temperature leads to several damaging processes, including unfolding and aggregation of cellular proteins. To avoid the disruption of protein homeostasis (proteostasis), all living organisms have evolved a fundamental protective mechanism known as heat-shock response [1]. Basically, it consists of a transcriptional reprogramming of cellular transcriptome, mainly finalized to the accumulation of protective proteins, collectively named heat-shock proteins (HSP), which counteract the deleterious effects of heat-shock on cellular polypeptides [2]. In detail, HSP mainly include molecular chaperones, which bind, stabilize and refold heat-damaged polypeptides, and proteases, that get rid of deleterious intracellular protein aggregates [3,4]. The heat-shock response relies on sophisticated regulatory mechanisms, which guarantee a precise control of HSP amount inside the cell. Indeed, under normal growth conditions, HSP must be expressed at a basal level to fulfill their role in promoting folding of newly synthetized polypeptides. On the other hand, upon a temperature upshift, HSP expression is strongly boosted for a limited period of time, during which they accumulate in the cytoplasm and carry out their protective role. Even though the heat-shock response constitutes a universal protective process, conserved in all kingdoms of life, the molecular mechanisms controlling HSP expression vary significantly even among different bacterial species. Prokaryotic microorganisms, in fact, can adopt positive and negative transcriptional regulators, often in combination with posttranscriptional mechanisms, to rapidly accumulate HSP only when they are required [5].

*Campylobacter jejuni* is a Gram-negative, curved, slender, motile-rod and is found in food, feces, and water. It represents an enteric pathogen that is associated with diarrhea and enterocolitis in humans and many animal species [6]. Although most people infected with *C. jejuni* spontaneously recover completely within a week, in rare cases infection results in long-term health problems [7]. It has been estimated that 5–20% of people with *C. jejuni* infection develop irritable bowel syndrome, 1–5% develop arthritis and one in every 1,000 reported *C. jejuni* illnesses leads to Guillain-Barré syndrome, an acute autoimmune neurological disorder that causes progressive muscle weakness and partial paralysis [8,9]. Recalling the ability of the pathogen to colonize different hosts and to adapt to diverse environments (animal hosts with different body temperatures as well as external environments), it has been hypothesized that temperature could represent an activation signal for host infection, and, in turn, for the heat-shock response, which would contribute to the switch between commensalism and pathogenicity [10,11]. In *C. jejuni*, the heat-shock response is orchestrated by the concerted action of two transcriptional repressors, which underpin a negative strategy of regulation of HSP expression [12,13]. As schematically drawn in Figure 1, the principal HSP of *C. jejuni* are grouped in four multicistronic operons, which harbor also the genes coding for the homologs of two heat-shock transcriptional repressors, named HspR and HrcA, widely distributed among bacterial species [14,15].

HspR plays the role of the master repressor of the regulatory circuit. At temperatures up to 37 °C, HspR binds and represses its own promoter (P*cbp*), as well as the regulatory region controlling the transcription of the *clpB* containing operon (P*clp*). Conversely, on the promoters upstream of the *groES-groEL* and *hrcA-grpE-dnaK* operons, HspR combines with HrcA and together exert transcriptional repression. In addition, it has recently been demonstrated a direct protein–protein interaction between HrcA and HspR and it has been shown that their binding on co-regulated targets occurs in a cooperative manner [16]. This negative regulatory circuit implies, at 37 °C which represents the normal temperature for the human host, a transcriptional repression of HSP-encoding genes operated by HspR and HrcA repressors bound to their operators within the heat-shock promoters. Upon thermal stress, this signal is in some way perceived by the regulatory system, the HspR and HrcA repressors are expected to detach from their binding sites, leading to transcriptional de-repression of heat-shock genes. Indeed, it has been demonstrated through whole transcriptomic analysis that a heat-shock treatment of *C. jejuni* cells at 42 °C for a limited period of time is sufficient to provoke a strong upregulation of several genes, including the operons controlled by HspR and HrcA [17].

In order to promptly respond to thermal stress, the environmental signal must be detected and transduced to the regulatory system, resulting in a coordinated gene expression output. In this respect, there are different mechanisms of thermal sensing which involve nearly all classes of biological molecules, including DNA, RNA, proteins and lipids [5,18]. More specifically, the perception of thermal stress can be indirect or direct. Indirect heat-sensing is based on the detection of the consequences of a sudden temperature increase, such as the intracellular accumulation of misfolded proteins. Examples of indirect heat-sensing involve chaperones as first-line detectors of stress: the increased amount of misfolded proteins in the cytoplasm titrates chaperones like DnaK and GroE, which are no more available to positively or negatively modulate the DNA-binding activity of heat-shock transcriptional regulators (like *Escherichia coli* σ^32^ interacting with DnaK and GroE, or HrcA/HspR repressors modulated by GroE and DnaK, respectively, in some bacterial species) [19,20,21,22,23,24]. On the other hand, direct heat sensing is based on the fact that temperature directly affects the activity of the sensing biomolecule. RNA molecules coding for heat-shock proteins, but also for virulence factors, can in some cases act as direct heat sensors. At lower temperatures, these transcripts assume a peculiar structure at their 5′-end that masks sequence determinants important for translation initiation, thereby impairing this process. Upon temperature increase, the inhibitory structure melts and translation of the mRNA is enhanced [25]. In a limited number of cases, it has been demonstrated that also heat-shock transcriptional regulators themselves can act as intrinsic heat-sensors, which are able to modulate target genes’ transcription in response to temperature changes. In detail, these heat-sensing regulators are competent for DNA-binding and can control gene transcription only at physiological temperature. Upon a sudden temperature increase, they go through a temperature-induced conformational rearrangement that lowers relative binding affinity for their binding sites, thereby influencing target gene expression. It has been reported that this kind of temperature-sensing system triggers heat-shock response in *Bacillus subtilis*, *Streptomyces albus*, *Salmonella enterica*, *Yersinia pestis* and *Helicobacter pylori* [26,27,28,29,30].

Recently, we determined the *C. jejuni* HrcA and HspR interactions on heat shock promoters by high-resolution DNase I footprints, showing that while DNA-binding of HrcA covers a compact region, HspR interacts with multiple high- and low-affinity binding sites [16]. In the present work, we show for the first time that in *C. jejuni* the HrcA repressor is the intrinsic heat-sensor of the heat-shock regulatory circuit. The results here presented suggest that a slight temperature increase irreversibly affects HrcA binding capacity to DNA, without influencing the partner repressor HspR. Moreover, we also report that HrcA binding activity is positively modulated by the chaperonin GroE, upon direct protein–protein interaction. The implications of these findings and a possible model for *C. jejuni* thermoregulation of heat-shock genes are discussed.

## 2. Materials and Methods

### 2.1. Bacterial Strains and Culture Conditions

The *Escherichia coli* strain BL21(DE3) was used for the overexpression of recombinant proteins, while DH5α strain was used for routine plasmid preparation and for cloning purposes (Appendix A). *E. coli* cells were grown in Luria-Bertani (LB) medium and when required, ampicillin was added to a final concentration of 100 µg/mL. *C. jejuni* NCTC 11168 cells were revitalized from glycerol stocks on Brucella broth agar plates containing 5% fetal calf serum, under microaerophilic conditions at 37 °C and 95% humidity in a water jacketed incubator (Thermo Scientific, Waltham, MA, USA). Liquid cultures were grown in Brucella Broth supplemented with 5% fetal calf serum with gentle agitation (120 rpm), under microaerophilic conditions (Oxoid, Basingstoke, Hampshire, UK).

### 2.2. Molecular Biology Procedures

Common molecular biology procedures including plasmid DNA transformation, restriction and amplification were performed as previously described by Sambrook et al. [31]. All modification enzymes and restriction endonucleases were used according to the manufacturer’s instructions (New England Biolabs, Ipswich, MA, USA). Mini- and midi-scale plasmid preparations were obtained using the NucleoSpin plasmid and the NucleoBond Xtra Midi plasmid purification kits, respectively (Macherey-Nagel GmbH & Co., KG, Düren, Germany). Following PCR amplification, DNA fragments were purified with the NucleoSpin Gel and PCR Clean-up kit (Macherey-Nagel GmbH & Co, KG, Düren, Germany). PCR reaction was carried out in an Applied Biosystems SimpliAmp thermal cycler (Thermo Scientific, Waltham, MA, USA) using PCRBIO Classic Taq and PCRBIO HiFi polymerase (PCR Biosystems Ltd., London, United Kingdom) on *C. jejuni* NCTC 11168 genomic DNA template. All plasmids and primers used in this work are listed in Appendix A.

### 2.3. Purification of Recombinant Proteins

*C. jejuni* recombinant His-tagged HrcA and HspR proteins were expressed in *E. coli* BL21(DE3) and purified through Ni^2+^-NTA affinity chromatography as previously described [16]. At the end of the purification, recombinant HrcA and HspR were stored in aliquots at −80 °C in Storage Buffer (for HrcA: 10 mM Tris-HCl, pH 8.0; 300 mM NaCl; 5 mM TCEP; 0.05% NP40; 10% glycerol; for HspR: 50 mM Tris-HCl, pH 8.0; 300 mM NaCl; 1 mM DTT; 0.05% NP40; 10% glycerol). A similar approach was used for the purification of GroEL and GroES proteins. In detail, the *groEL* and *groES* genes were PCR-amplified from *C. jejuni* NCTC 11168 genome, cloned into pET15b and the resulting pET15b-*groEL*/pET15b-*groES* constructs were separately transformed in *E. coli* BL21(DE3) strain. Bacterial cells harboring the desired plasmid were inoculated in 400 mL of LB medium supplemented with ampicillin and grown at 37 °C to an OD_600_ of 0.5. Then, 0.5 mM isopropyl-β-D-thiogalactopyranoside was added to induce recombinant protein expression and incubation was continued for 4 h at 37 °C. After harvesting cells by centrifugation, the pellet was resuspended in 12 mL of Lysis Buffer (50 mM NaH_2_PO_4_; 300 mM NaCl; 10 mM imidazole; 10% glycerol; pH 8.0) containing 1 mg/mL lysozyme and disrupted by sonication. Following the removal of cellular debris by centrifugation, the soluble fraction was incubated with 300 μL of 50% Ni^2+^-NTA slurry (Merck KGaA, St. Louis, MO, USA) for 1 h at 4 °C. Nonspecific proteins were removed by washing the slurry 5 times with Lysis Buffer and 4 times with Wash Buffer (50 mM NaH_2_PO_4_; 300 mM NaCl; 20 mM imidazole; 10% glycerol; pH 8.0). Then, the elution step was carried out with Elution Buffer (50 mM NaH_2_PO_4_; 300 mM NaCl; 250 mM imidazole; 10% glycerol; pH 8.0) and the recovered protein sample was dialyzed against two changes of Storage Buffer (the same buffer used for HrcA and HspR storage, see above). Recombinant protein preparations were analyzed by SDS-PAGE and quantified by Bradford colorimetric assay (BioRad, Hercules, CA, USA).

### 2.4. Electrophoretic Mobility Shift Assay (EMSA)

The region encompassing the *groES-groEL* (P*gro*) and *cbpA-hspR* (P*cbp*) promoters were PCR amplified with oligonucleotides CjPgro-F/R and CjPcbp-F/R (Appendix A), respectively, from *C. jejuni* genomic DNA. For EMSA assays, 10 ng of the DNA probe of interest were mixed with recombinant purified proteins in 20 µL of 1X FPBE buffer (10 mM Tris-HCl, pH 8.0; 60 mM NaCl; 10 mM KCl; 5 mM MgCl_2_; 1 mM DTT; 0.05% NP40-igepal; 10% glycerol) and incubated for different periods of times and temperatures (see Section 3.1, Section 3.2, Section 3.3 and Section 3.4). Then, binding reactions were separated in a 5.5% acrylamide/bisacrylamide (19:1) gel, in 0.5X TB (45 mM Tris-borate) running buffer, run at room temperature for 120 min at 100 V. Gels were stained in 0.5X TB containing 0.5 µg/mL Ethidium Bromide for 30 min followed by destaining in ultrapure water.

### 2.5. DNase I Footprinting

The *groES-groEL* and *cbpA-hspR* promoter regions were amplified by PCR from *C. jejuni* NCTC 11168 genomic DNA and cloned into pGEM-T-Easy plasmid (Promega, Madison, WI, USA; Appendix A). Radioactive labeled dsDNA probes were prepared as previously described [32]. Briefly, the plasmid containing the DNA region of interest (2 pmol) was linearized by enzymatic restriction with NcoI or NdeI endonucleases, dephosphorylated with Calf Intestinal Phosphatase enzyme and 5′-end labeled using [γ-^32^P]-ATP (5 pmol) with T4 Polynucleotide Kinase. DNA probe labeled at one extremity was separated from the plasmid backbone by a second enzymatic restriction and purified by preparative polyacrylamide gel electrophoresis. Protein–DNA complexes were allowed to form in 50 µL of Footprinting Buffer (5 mM Tris-HCl, pH 8.0; 60 mM NaCl; 5 mM KCl; 5 mM MgCl_2_; 0.2 mM DTT; 0.01% NP40-igepal and 10% glycerol), containing approximately 20 fmol of the labeled probe, 200 ng of sonicated salmon sperm DNA as nonspecific competitor, and increasing concentrations of recombinant protein(s). The reactions were incubated for different periods of times and temperatures (see Section 3.1, Section 3.2, Section 3.3 and Section 3.4), before adding 0.066 U of DNase I, prepared in Footprinting Buffer containing 5 mM CaCl_2_. Following 75 s of incubation at room temperature, the reactions were stopped by adding 140 μL of DNase I Stop Buffer (192 mM NaOAc, pH 5.2; 32 mM EDTA; 0.1% SDS; 64 μg/μL sonicated salmon sperm DNA). Samples containing the labeled DNA fragments were then purified and loaded on 8M urea–6% polyacrylamide (19:1) gels as previously described [16]. The gel was blotted onto a 3 MM Whatman paper sheet (Thermo Scientific, Waltham, MA, USA), dried and autoradiographed.

### 2.6. GST-Pulldown Assay

To carry out GST (glutathione S-transferase)-pulldown assay, *C. jejuni* NCTC11168 HrcA repressor (bait protein) was overexpressed as a fusion protein with GST. As control bait, the GST protein alone was overexpressed under the same experimental conditions as GST-HrcA. Protein expression was performed in *E. coli* BL21(DE3) transformed with the plasmids pGEX_NN_ (for GST alone) or pGEX_NN_-*hrcA* (for GST-HrcA) (Appendix A), following the experimental conditions previously described [33]. Then, each pellet from 250 mL bacterial cultures was resuspended in 10 mL of 1X phosphate-buffered saline (PBS) containing 1 mg/mL lysozyme and incubated for 60 min at 4 °C on a tilt-roll. Following the addition of DTT (5 mM final concentration), cells were sonicated on ice and centrifuged at maximum speed for 20 min. The resulting supernatant (soluble cell extract) was supplemented with 1% Triton X-100 and mixed with 160 μL of glutathione-agarose slurry (Merck KGaA, St. Louis, MO, USA), followed by incubation at 4 °C for 90 min. Nonspecific proteins were removed by washing the slurry 5 times with 3 mL of 1X PBS. Then, 200 μL of 1X PBS was added to the GSH-Sepharose slurry with bound GST and GST-HrcA. The amount of GST and GST-HrcA proteins bound to slurry was monitored by SDS-PAGE. A cytoplasmic protein extract from *C. jejuni* NCTC11168 wild type bacteria was obtained from a 40 mL liquid culture. In detail, bacterial pellet was resuspended in 1.5 mL of Res buffer (10 mM Tris-HCl, pH 7.5; 100 mM NaCl; 20 mM KCl; 1 mM EDTA; 0.01% NP-40 Igepal) containing 1 mg/mL lysozyme, sonicated and centrifuged. The whole volume of supernatant obtained was pre-cleared by mixing it with 50 μL of Glutathione-agarose slurry bound to GST and incubated for 90 min at 4 °C. After that, the sample was passed through an empty column which retained the resin, and the cleared extract was collected for the pulldown experiment. Then, the cleared sample was divided into two 700 μL aliquots, mixed with GST and GST-HrcA-Glutathione-agarose slurry (the same molar concentration of the two bait proteins—GST or GST-HrcA—was used) and incubated at 4 °C for 16 h. After that, the slurry was recovered by centrifugation (800× *g*, 1 min at 4 °C) and washed 4 times with 100 μL of Res buffer. Then, 1 bed volume of 1X SDS-PAGE loading buffer (Laemmli buffer, [31]) was added to the slurries, and samples were boiled for 10 min to detach all the proteins recovered from the resin. For immunoblot assay, the same volumes of samples from the GST and GST-HrcA columns were separated through a 12% SDS-polyacrylamide gel electrophoresis together with 50 ng of purified recombinant GroEL protein as positive control of the assay. After electrophoretic separation, proteins were transferred to a PVDF membrane. After blocking the membrane with 1X PBS containing 5% low-fat milk and 0.05% Tween-20, it was incubated overnight at 4 °C with a 1:2500 dilution of rabbit polyclonal primary antibody (anti-GroEL). After extensive washes in 1X PBS containing 0.05% Tween-20 (PBST), the membrane was incubated for 1 h at room temperature with a 1:5000-diluted peroxidase-conjugated goat anti-rabbit antibody (Invitrogen, Waltham, MA, USA). Following an additional washing step in PBST, the membrane was developed by pouring on it a solution of 1.25 mM luminol containing 0.015% H_2_O_2_ and 0.068 mM p-coumaric acid.

## 3. Results

### 3.1. The Heat-Shock Repressor HrcA Binds to Pgro Promoter in a Temperature-Dependent Manner

To assess if temperature can affect DNA-binding capabilities of the heat-shock repressor HrcA, in vitro protein–DNA interaction assays were set up. First of all, we carried out EMSA (Electrophoretic Mobility Shift Assay) experiments using P*gro* promoter probe and the purified HrcA protein. In order to investigate if temperature is able to influence HrcA activity, DNA-binding of the protein was promoted by incubating the reactions at 25 °C for 10 min (a condition previously used to study HrcA and HspR interaction with the DNA in [16]), then samples were shifted at 25, 37 or 42 °C for an additional period of time of 10 min, before separation of the complexes through native polyacrylamide gel electrophoresis. As reported in Figure 2A, at 25 °C the inclusion of increasing amounts of HrcA provoked the appearance of a single retarded band in a dose-dependent manner (Figure 2A, lanes 1–4), indicative of HrcA binding to P*gro*, which harbors a single DNA-binding site [16]. An identical result was obtained by incubating the DNA-binding reactions at 37 °C (Figure 2A, lanes 5–7). Interestingly, when the reaction mixes were incubated for 10 min at 42 °C, the intensity of the retarded band dropped at all the concentrations tested (Figure 2A, lanes 8–10), indicating a significant loss of HrcA DNA-binding activity at this temperature.

To further characterize HrcA behavior on P*gro* promoter at different binding temperatures, we carried out DNase I footprinting assays with purified HrcA on P*gro* labeled probe, following the same experimental layout used for EMSA. Briefly, protein–DNA complexes were allowed to form at 25 °C, then reaction mixes were shifted to different temperatures (25, 37 or 42 °C) for 10 min before DNase I digestion. As shown in Figure 2B, upon the addition of increasing concentrations of protein to the reaction at 25 °C, an area of protection indicative of HrcA binding to the P*gro* probe (marked by boxes) appeared (Figure 2B, lanes 1–4). In the same experiment, diverse DNase I hypersensitive sites were detected (marked by arrowheads). Likely, these arise from a conformational change of the probe following protein–DNA interaction. An almost identical binding pattern was obtained with the same set of reactions incubated at 37 °C (Figure 2B, lanes 5–8). On the contrary, the same experiment performed at 42 °C showed a complete reduction of HrcA DNA-binding capabilities. At this incubation temperature, protected regions as well as DNase I hypersensitive sites disappeared completely (Figure 2B, lanes 9–12). Thus, HrcA binding to P*gro* promoter, at least in vitro, is temperature-dependent, and the repressor loses its DNA-binding activity at 42 °C.

### 3.2. The Heat-Shock Master Regulator HspR Is Not Sensitive to Temperature Changes

Recalling the fact that in *C. jejuni* some heat-shock induced promoters (specifically P*gro* and P*hrc*) are co-repressed by the cooperative action of HrcA and HspR [16,17], we decided to assess if this latter regulator displays a DNA-binding behavior similar to HrcA in response to temperature changes.

To this aim, we expressed and purified the HspR recombinant protein and used it to carry out DNA-binding assays at different temperatures. The experimental set up of HspR DNA-binding assays adhered to the same principles as for HrcA, consisting of an incubation period of 10 min at 25 °C followed by an additional 10-min step at 25, 37 or 42 °C. Figure 3A shows the result of a representative EMSA experiment carried out as described.

When increasing amounts of HspR were incubated with P*gro* at 25 °C, several shifted bands, characterized by different electrophoretic mobilities, appeared. At low protein concentration, two retarded bands (Figure 3A, lanes 2 and 3, indicated by * and ** symbols) became evident, while a third shifted band appeared only at the highest HspR concentration tested (Figure 3A, lane 4, indicated by *** symbol). This result is consistent with the presence of three flanking HspR operators on P*gro* promoter, as previously defined [16]. Interestingly, we obtained an identical result when we carried out DNA-binding experiments at higher temperatures. As clearly evident in Figure 3A, the pattern of retarded bands obtained in EMSA for DNA–protein complexes incubated at 37 °C (Figure 3A, lanes 5–7) and at 42 °C (Figure 3A, lanes 8–10) were almost identical to the result of the control experiment at 25 °C (Figure 3A, lanes 2–4), suggesting that HspR maintained its DNA-binding activity across the range of temperatures tested. To further corroborate these results, we also analyzed HspR interaction with P*gro* promoter through DNase I footprinting experiments (Figure 3B). Upon incubation of the labeled probe with increasing amounts of HspR at 25 °C, regions of protection and sites of DNase I-hypersensitivity appeared, marking the HspR binding region on P*gro* promoter previously mapped (Figure 3B, lanes 1–3) [16]. The incubation of DNA–protein complexes at 37 and 42 °C provoked the same binding pattern obtained at 25 °C (Figure 3B, lanes 4–6 and 7–9), sustaining the hypothesis that, in the experimental condition tested, the DNA-binding activity of the HspR repressor to P*gro* is not influenced by temperature.

Next, we wanted to assess the temperature-dependent DNA-binding behavior of HspR on promoters exclusively controlled by this master repressor. To this aim, we selected the P*cbp* promoter region, exclusively bound and repressed by HspR [16], as probe to perform DNA-binding assays with purified HspR at different temperatures (Figure 4). Specifically, EMSA assay carried out at different temperatures, (same conditions as in Figure 3A), demonstrated that HspR on P*cbp* promoter behaved in the same way as on P*gro* at different temperatures (Figure 4A). Inspecting more in detail the EMSA on the P*cbp* promoter (Figure 4A), at all the temperatures tested, increasing amounts of HspR added to the binding reaction led to the appearance of two bands with lower electrophoretic mobility, compared to the free probe, consistent with the presence of two distinct binding sites for the repressor on this promoter [16]. Results obtained indicate that HspR maintains the same binding capabilities to P*cbp* at the different temperatures tested, evoking the observations made for the P*gro* promoter (Figure 3).

Furthermore, DNase I footprinting analysis on labeled P*cbp*, carried out following the same experimental layout as described above, confirmed the data obtained using EMSA approach (Figure 4B). In this assay, the addition of increasing concentrations of HspR enhanced the appearance of two regions of protection and two DNase I hypersensitive bands (Figure 4B, marked by black boxes and arrowheads on the right side, respectively). In addition, the HspR binding pattern obtained was comparable among the different temperatures of incubation. Taken together, all these data indicate HspR as a regulatory protein whose DNA-binding activity is not affected by temperature variations, at least in the range tested.

*C. jejuni* can colonize (as commensal as well as a pathogen) different hosts with diverse body temperatures and can experience heat-stress conditions of higher severity [34], beyond the typical 42 °C commonly chosen to mimic heat-shock conditions. For this reason, we hypothesized that HspR, which appears to be rather stable even at 42 °C, could be a sensor of higher environmental temperatures. To explore this possibility, we assayed HspR binding capabilities to P*cbp* at 45 °C, in comparison to 37 and 42 °C through EMSA assay (Appendix A), which demonstrated that HspR binding is unaffected even at this temperature.

Considering that on the P*gro* promoter, both proteins are necessary to establish repression [13,16], it can be hypothesized that on dual controlled promoters HrcA represents the specific temperature sensor, while HspR loses its DNA contacts as a consequence of HrcA temperature-dependent behavior. In light of this consideration, we assessed the DNA-binding behavior of HrcA and HspR to P*gro* at different temperatures, including a reaction in which both proteins were simultaneously incubated with the DNA probe. Appendix A shows that, while at 25 °C the inclusion of both proteins provoked the appearance of low electrophoretic mobility bands, indicative of the simultaneous interaction of HrcA and HspR with the probe, at 42 °C the pattern of retarded bands resulting from the inclusion of HrcA and HspR appeared almost identical to the one obtained upon incubation of HspR alone (Appendix A). These in vitro results suggest that on the co-regulated P*gro* promoter, HspR binding is unaffected at 42 °C regardless the loss of binding of the partner repressor HrcA.

### 3.3. The DNA-Binding Activity of HrcA Is Irreversibly Lost upon Heat Challenge

Most of transcriptional regulators, whose activity is modulated by temperature, go through a reversible structural transition and are able to recover their DNA-binding capability when permissive conditions are restored. To assess if the loss of HrcA DNA-binding activity following heat-treatment is a reversible process, we modified the heat-treatment protocol used in in vitro binding assays. In detail, the recombinant HrcA protein was incubated with the P*gro* probe at 25 °C for 10 min to allow the formation of protein–DNA complex, then heat-treated at different temperatures (as before, 25, 37 or 42 °C) for 10 min. Following this heat-challenge profile, the temperature was lowered at 25 °C for a 10-min recovery step before electrophoretic separation (EMSA assay). As shown in Figure 5A, loss of HrcA DNA-binding activity upon exposure of the protein at 42 °C could not be recovered by restoring the incubation temperature at a permissive value. Indeed, when the protein–DNA reactions were incubated at 42 °C before recovery at 25 °C, the intensity of the retarded band dropped at all the concentrations tested (Figure 5A, lanes 8–10), compared to the intensity of the shifted band obtained following incubation at lower temperatures (Figure 5A, lanes 2–4 and 5–7). To further confirm this observation, we carried out DNase I footprinting assay in which the labeled probe was digested following the HrcA heat treatment just described for the EMSA assay (in this case the temperature challenge was limited to 25 and 42 °C). As clearly evident in Figure 5B, the HrcA binding pattern dramatically weakens upon incubation of the protein at 42 °C, regardless of the recovery step at 25 °C (Figure 5A, lanes 5–8 compared to lanes 1–4). Taken together, these protein–DNA interaction experiments unambiguously show that, in the absence of other cellular factors, the DNA-binding activity of HrcA is irreversibly affected by heating in vitro.

### 3.4. The Chaperonin GroE Stimulates HrcA Binding to DNA 

The data presented above (Section 3.3) portray HrcA as a protein highly sensitive to heat fluctuations, whose DNA-binding activity is dramatically affected by slight increases of few degrees in reaction temperature. In addition, following HrcA affinity purification, we observed that the stability of the protein in solution was quite poor, showing the tendency to form aggregates [16], a behavior previously reported for the homologous HrcA proteins of other organisms [23,35,36]. Proteins characterized by such low-stability features are frequently assisted by cellular chaperones, which promote the acquisition and the maintenance of the correct folding. Furthermore, it is well-documented that, in some other bacterial species, the DNA-binding activity of transcriptional regulators involved in controlling heat-shock response is modulated by chaperones that they control [5]. All together, these considerations prompted us to explore the possibility of a functional interaction between HrcA and the chaperonin GroE. To assess if the chaperonin GroE is able to modulate HrcA activity, we set up a DNase I footprinting analysis, in which we compared DNA-binding affinity of the repressor to P*gro* promoter in the presence or absence of GroEL and of the co-chaperonin GroES (Figure 6).

In the absence of GroE in the reaction, clear areas of protection of the probe were obtained only at the highest HrcA concentration used, interspersed with several DNase I hypersensitive sites (Figure 6, lane 5). Interestingly, the inclusion of a complete GroESL chaperonin system in the binding reactions led to an enhancement of HrcA interaction with P*gro*. As it is possible to appreciate in Figure 6, HrcA binding pattern (i.e., protected regions and sites of hypersensitivity to DNase I digestion) became clearly detectable at a lower protein concentration (Figure 6, lanes 8–10), indicative of an increase in the affinity of HrcA for its operator. Furthermore, the same conclusions were drawn analyzing GroE effect on HrcA DNA-binding activity through EMSA (Appendix A), which confirmed a chaperonin-dependent stimulation of HrcA binding to P*gro* promoter. In conclusion, these experiments suggest that GroE is able to enhance HrcA DNA binding affinity for its binding site on P*gro*, thereby contributing to the transcriptional repression of the HrcA controlled promoters.

To experimentally test the interaction between the HrcA repressor and GroE chaperonin, we carried out a GST-pulldown assay. To this aim, we expressed and purified the recombinant GST-HrcA fusion protein, which was used as bait and incubated with a *C. jejuni* total protein extract. A recombinant GST protein was used as a control bait and incubated with an identical *C. jejuni* protein extract. Following the recovery of the bait proteins (GST alone and GST-HrcA) associated with their interactor(s) from a glutathione-agarose slurry, we monitored the presence of GroEL in the obtained samples by carrying out an immunoblot assay stained with a specific (α-GroEL) antibody (Figure 7).

When we used the GST-HrcA protein as bait, a band corresponding to the expected molecular weight of GroEL (60 kDa) was detected (Figure 7, lane labeled “PD” in the GST-HrcA sample set), while this band was much less recovered in the control sample in which the GST bait was used (Figure 7, lane labeled “PD” in the GST sample set). Notably, a cross-reactive band at a lower molecular weight (just below 52 kDa, indicated by the symbol × in Figure 7), with similar intensity in all the samples analyzed, confirms the same recovery of background proteins at the end of the pulldown procedure, further sustaining the enrichment of the specific GroEL band observed. These data confirm the association between the *C. jejuni* heat shock repressor HrcA and the chaperonin GroE.

## 4. Discussion

In this work we aimed at characterizing the temperature sensitivity of the two transcriptional repressors HrcA and HspR, governing the heat-shock response in *C. jejuni*. Data here presented demonstrate a completely different behavior of these regulators upon an increase in the environmental temperature. Specifically, we observed that HrcA is sensitive to slight temperature upshift, as it irreversibly loses DNA-binding activity following an increase of reaction temperature from 37 to 42 °C (Figure 2 and Figure 5). On the contrary, in the experimental conditions used and temperature range tested, HspR behaved as a stable protein and its DNA-binding activity to its operators is unaffected by temperature changes (Figure 3, Figure 4 and Appendix A), both on a promoter co-regulated in association with HrcA (P*gro*, Figure 3), as well as on an exclusively HspR-dependent promoter (P*cbp,* Figure 4). Even though we cannot exclude the hypothesis that HspR could perceive temperature stress of higher severity (in our experimental set up it appeared not sensitive to temperatures up to 45 °C, Appendix A), our data would suggest HspR heat detection may indirectly be mediated by a yet unidentified protein partner.

The interplay between HrcA and HspR, on which heat-shock regulation relies in *C. jejuni*, is based on the dual control of P*gro* and P*hrc* promoters exerted by both repressors (Figure 1). On these regulatory regions, binding of HrcA takes place in the core promoter region, while HspR operators map upstream of the core promoter elements -10 and -35 [16]. Thus, loss of HrcA binding and promoter occupancy upon temperature increase could be sufficient to remove repression and allow transcription initiation by RNA polymerase. Moreover, on co-regulated promoters, HrcA and HspR bind cooperatively to their respective operators and directly interact [16]. Even though in vitro EMSA experiment in which we assessed the influence of temperature on DNA-binding behavior of HrcA and HspR simultaneously incubated to P*gro* suggests the absence of any effect of temperature-responsive HrcA on HspR (Appendix A), it cannot be excluded that in vivo, upon a temperature challenge, loss of HrcA binding affinity is sufficient to destabilize the whole repressive complex.

According to our observations, HrcA represents an example of intrinsic heat-sensing transcriptional regulator. Among the different ways in which temperature variations can be perceived, the use of intrinsic heat-sensing protein thermometers has been observed both in Gram-negative and Gram-positive species [5]. However, so far, a restricted number of bacterial transcriptional regulators displaying this changing behavior in response to temperature variations has been experimentally characterized [26,27,28,29,30,37]. Notably, for most temperature-sensitive transcriptional regulators studied so far, the temperature increase provokes a reversible conformational change, which is reversed once the temperature returns to a permissive level. In these examples, the temperature-induced conformational transition can affect the functional oligomeric state of the protein (as it happens for RheA of *Streptomyces albus* [26] and TlpA of *Salmonella enterica* serovar Typhimurium [28]) or the orientation of a small loop belonging to the DNA-binding domain (like the CtsR repressor of low-GC Gram-positive bacteria, in which temperature influences the position of a glycine-rich loop within the winged helix–turn–helix DNA binding domain [27]). A completely different scenario has been observed in *H. pylori*, where upon a temperature upshift to 42 °C, the HrcA repressor goes through a major structural transition, an event that irreversibly compromises its DNA-binding activity [30]. In this work, we show that, in the absence of other cellular factors, the DNA-binding activity of *C. jejuni* HrcA is irreversibly affected by heating (Figure 5). This common behavior of the two HrcA proteins from *H. pylori* and *C. jejuni* is not surprising, as they share 27.3% of amino acids identity, are both unstable proteins, display a strong tendency to form aggregates in vitro when incubated at temperatures above 40 °C. Furthermore, the DNA-binding activity of *H. pylori* HrcA is stimulated by the GroE chaperonin under normal conditions of growth, as well as following a heat-challenge to 42 °C [30,38]. Although we do not have any structural data, it could be hypothesized that when exposed to non-permissive conditions *C. jejuni* HrcA, similarly to what happens for its homologous repressor of *H. pylori*, undergoes a structural unfolding. In that case, this peculiar behavior in response to temperature increase could represent a feature characterizing the HrcA family of heat-shock repressors.

Beyond the characterization of HrcA binding at different temperatures, in this work we show for the first time that the chaperonin GroE positively modulates HrcA activity (Figure 6 and Figure 8). This functional interaction between heat-shock repressors and chaperones has previously been observed in other bacterial species, including *Bacillus subtilis* and *Chlamydia trachomatis* [23,24]. Considering that we observed GroE-mediated stimulation at permissive temperature, we can put forward the hypothesis that, in analogy to the titration model initially proposed for *B. subtilis*, in *C. jejuni* the chaperonin GroE interacts with HrcA to assist its folding and enhance its DNA binding activity. Following a heat-stress, GroE would be sequestered by increasing amounts of misfolded proteins accumulating in the cytoplasm, relieving HrcA transcriptional repression of the heat-shock promoters. Considering the HrcA heat-shock induced loss of DNA binding activity presented here, a possibility is that GroE can assist also HrcA refolding, lost upon a heat-challenge, when permissive conditions are restored (Figure 8). To test this hypothesis, we assayed in vitro the ability of GroE to restore the DNA-binding activity of HrcA, lost upon exposure to 42 °C. The footprinting assay shown in Appendix A suggests that, in the experimental conditions used, the GroE chaperonin has a barely visible effect on HrcA functionality. However, we cannot rule out the hypothesis that in vivo, GroE takes part in the regulation of HrcA DNA-binding capability also upon a heat-challenge. In other words, following a heat-shock, when temperature returns to a permissive value, at least part of the HrcA protein could be refolded by GroE, thereby retrieving its DNA binding activity, while some unfolded repressor molecules would be degraded by dedicated proteases belonging to the protein quality control network of *C. jejuni*, as it happens for other heat-shock repressors [5] (Figure 8). When temperature shifts down again, this hypothetical fraction of GroE-refolded HrcA protein would contribute, together with the newly synthesized polypeptide, to restore the transcriptional repression of heat-shock genes.

## Figures and Tables

**Figure 1 biomolecules-11-01413-f001:**
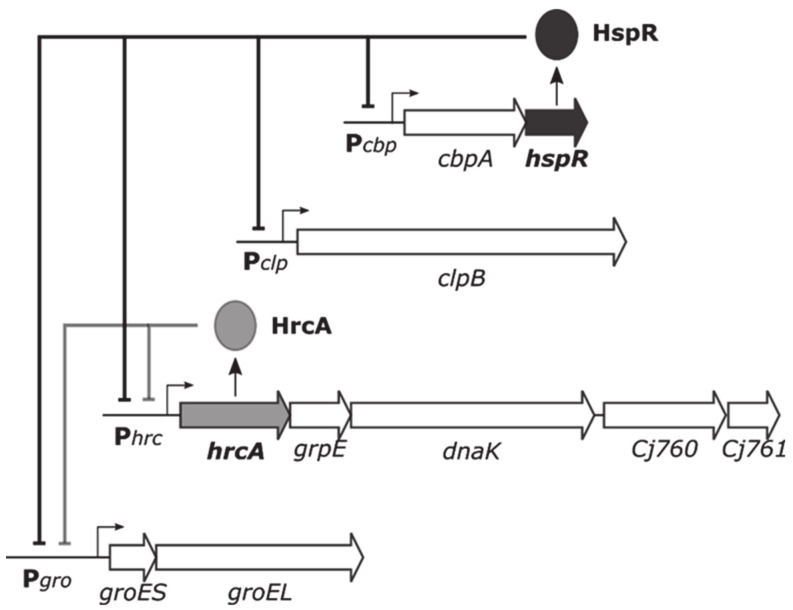
Schematic representation of the four heat shock operons containing the genes encoding the major *C. jejuni* HSP and their transcriptional regulation exerted by the combined action of HrcA and HspR repressors. Drawn according to previous results [16].

**Figure 2 biomolecules-11-01413-f002:**
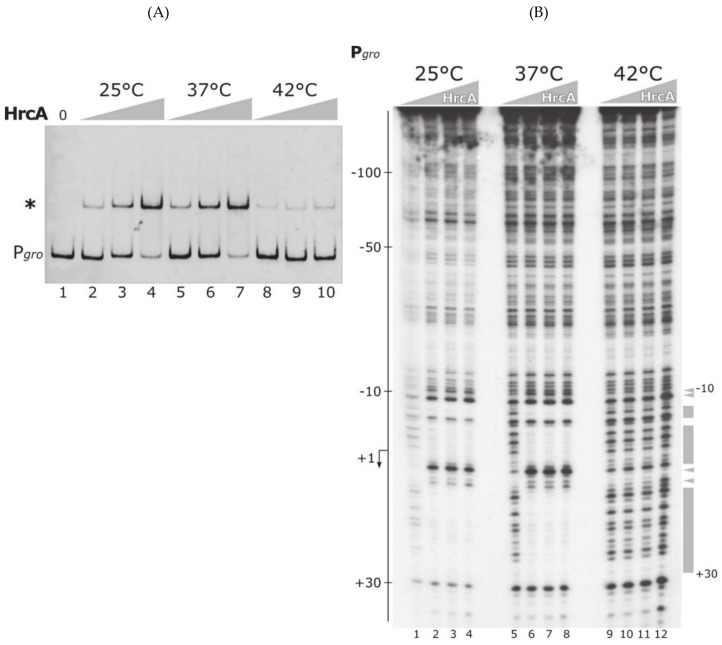
DNA-binding assays of the heat-shock repressor HrcA to the P*gro* promoter at different temperatures. (**A**) EMSA carried out with purified HrcA on P*gro* promoter probe. DNA–protein complexes were allowed to form for 10 min at 25 °C, then reactions were moved to different temperatures (25, 37 or 42 °C) for 10 min. An asterisk marks the HrcA shifted band, while the label “P*gro*” indicates the free probe. Lanes 1–4 contain 0, 45, 90 and 180 nM HrcA, respectively. Lanes 4–6 and 8–10 contain the same increasing concentrations of HrcA as in samples 2–4. Grey triangles are indicative of HrcA concentrations at the indicated temperatures. (**B**) DNase I footprinting assays with HrcA on P*gro* labeled probe at different temperatures. Protein–DNA mixes were incubated for 10 min at 25 °C, then moved to different temperatures (25, 37 or 42 °C) for 10 min, before DNase I cleavage. Lanes 1–4, 5–8 and 9–12 contain 0, 90, 180 and 360 nM HrcA, respectively. On the left of the autoradiograph, the numbers refer to the positions with respect to the transcriptional start site (indicated by a bent arrow). Protected regions and DNase I hypersensitive sites are indicated on the right by grey boxes and arrowheads, respectively, together with positions delimiting the HrcA binding site. HrcA concentrations are depicted on top of the figures with grey triangles at the indicated temperatures. Symbol * indicates the HrcA shifted band.

**Figure 3 biomolecules-11-01413-f003:**
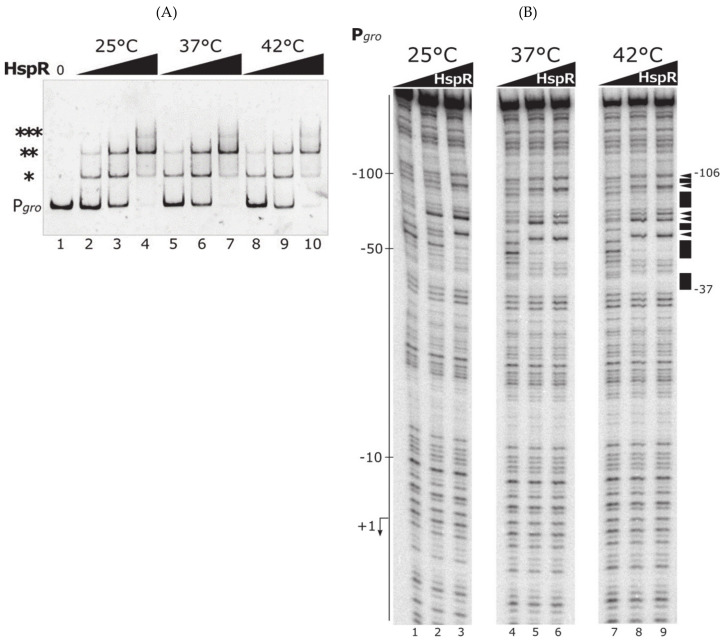
DNA-binding assays of the heat-shock master regulator HspR to the P*gro* promoter at different temperatures. (**A**) EMSA carried out with purified HspR on P*gro* promoter probe. DNA–protein complexes were allowed to form for 10 min at 25 °C, then reactions were shifted to different temperatures (25, 37 or 42 °C) for 10 min. On the left, asterisks mark the different HspR shifted bands, while the label “P*gro*” indicates the free probe. Lanes 1–4 contain 0, 15, 30 and 60 nM HspR, respectively. Lanes 4–6 and 8–10 contain the same increasing concentrations of HspR as in samples 2–4. Black triangles represent increasing concentrations of HspR at the indicated temperatures; 0, no protein added. (**B**) DNase I footprinting assays with HspR on P*gro* labeled probe at different temperatures. Protein–DNA mixes were incubated for 10 min at 25 °C, then moved to different temperatures (25, 37 or 42 °C) for 10 min, before DNase I cleavage. Lanes 1–3, 4–6 and 7–9 contain 0, 30 and 60 nM HspR, respectively. Black triangles are as in panel A. On the left of the autoradiograph, the numbers refer to the positions with respect to the transcriptional start site (indicated by a bent arrow). Protected regions and DNase I hypersensitive sites are indicated on the right by black boxes and arrowheads, respectively, together with positions delimiting the HspR binding site. Symbols *, ** and *** mark HspR shifted bands.

**Figure 4 biomolecules-11-01413-f004:**
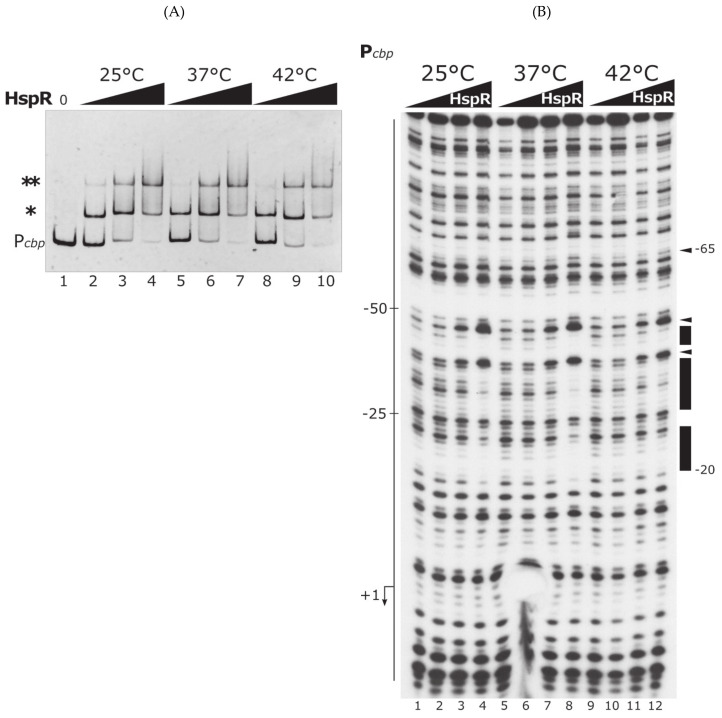
DNA-binding experiments of HspR to the P*cbp* promoter at different temperatures. (**A**) EMSA carried out with purified HspR on P*gro* promoter probe. DNA–protein complexes were allowed to form for 10 min at 25 °C, then reactions were moved to different temperatures (25, 37 or 42 °C) for 10 min. On the left, asterisks (* and **) mark the different HspR shifted bands, while the label “P*cbp*” indicates the free probe. Lanes 1–4 contain 0, 15, 30 and 60 nM HspR, respectively. Lanes 4–6 and 8–10 contain the same increasing concentrations of HspR as in samples 2–4. Black triangles indicate increasing amounts of HspR at the indicated temperatures; 0, no protein added. (**B**) DNase I footprinting assays with HspR on P*cbp* labeled probe at different temperatures. Protein–DNA mixes were incubated for 10 min at 25 °C, then shifted to different temperatures (25, 37 or 42 °C) for 10 min, before DNase I cleavage. Lanes 1–4, 5–8 and 9–12 contain 0, 30, 60 and 120 nM HspR, respectively. On the left of the autoradiograph, the numbers refer to the positions with respect to the transcriptional start site (indicated by a bent arrow). Protected regions and DNase I hypersensitive sites are indicated on the right by black boxes and arrowheads, respectively, together with positions delimiting the HspR binding site. Black triangles are as in panel A. Symbols * and ** mark HspR shifted bands.

**Figure 5 biomolecules-11-01413-f005:**
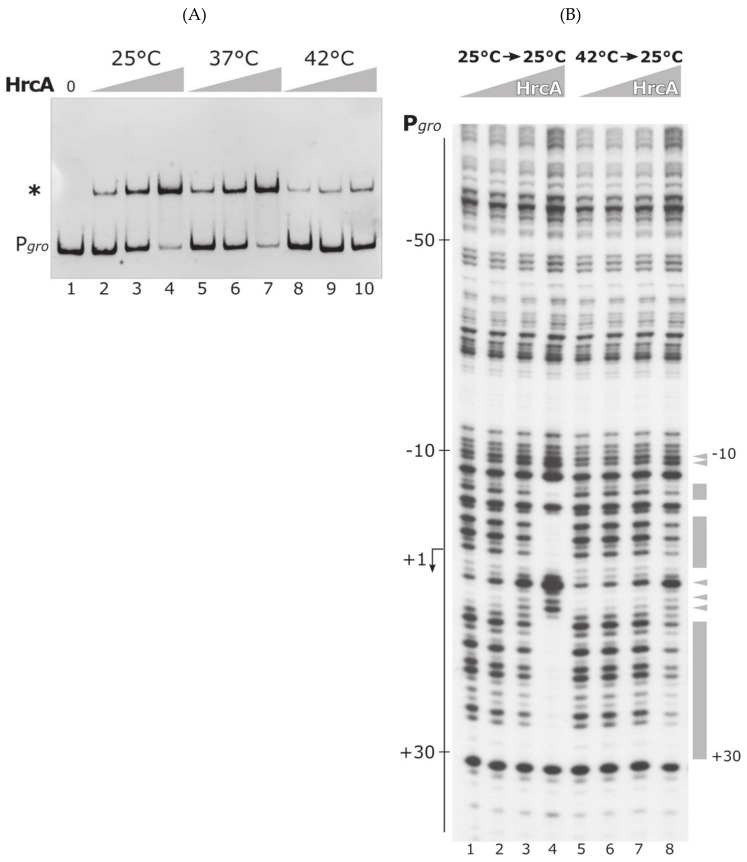
DNA-binding experiments of HrcA exposed to different temperatures and recovered at 25 °C. (**A**) EMSA carried out with purified HrcA on P*gro* promoter probe. DNA–protein complexes were allowed to form for 10 min at 25 °C, then reactions were moved to different temperatures (25, 37 or 42 °C) for 10 min. Following heat-challenge, reactions were incubated at 25 °C for a 10-min recovery step [26]. On the left, an asterisk marks the HrcA shifted band, while the label “P*gro*” indicates the free probe. Lanes 1–4 contain 0, 45, 90 and 180 nM HrcA, respectively. Lanes 4–6 and 8–10 contain the same increasing concentrations of HrcA as in samples 2–4. Grey triangles indicate increasing amounts of HspR at the indicated temperatures; 0, no protein added. (**B**) DNase I footprinting assays with HrcA on P*gro* labeled probe. Protein–DNA mixes were incubated for 10 min at 25 °C, then moved to 25 or 42 °C for 10 min, followed by a recovery step at permissive (25 °C) temperature, before DNase I cleavage. Lanes 1–4 and 5–8 contain 0, 45, 90 and 180 nM HrcA, respectively. On the left of the autoradiograph, the numbers refer to the positions with respect to the transcriptional start site (indicated by a bent arrow). Protected regions and DNase I hypersensitive sites are indicated on the right by grey boxes and arrowheads, respectively, together with positions delimiting the HrcA binding site. Grey triangles are as in panel a. Symbol * indicates the HrcA shifted band.

**Figure 6 biomolecules-11-01413-f006:**
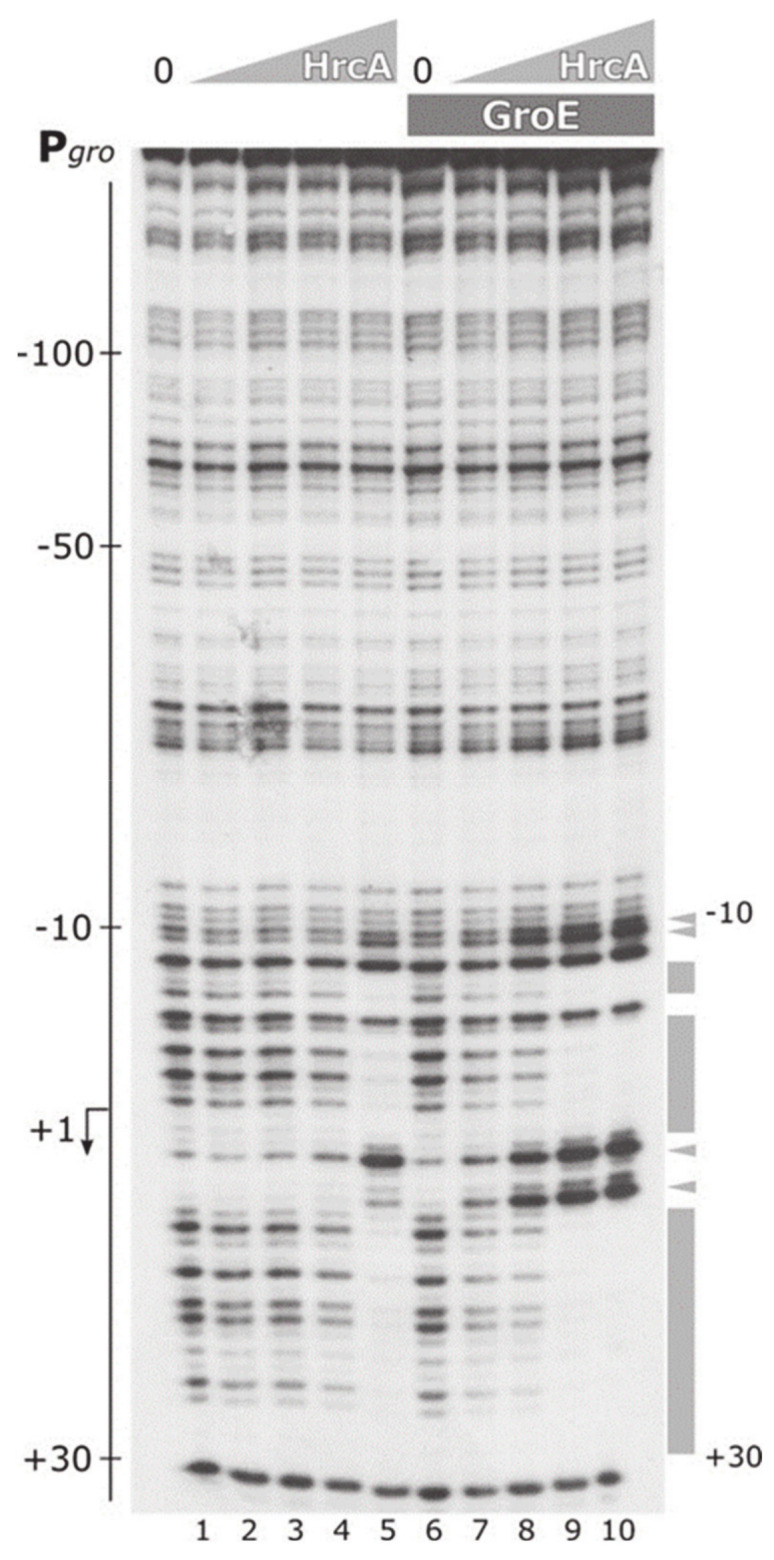
DNase I footprinting analysis of HrcA binding to P*gro* promoter in the presence or absence of GroEL/GroES complex. Protein–DNA mixes were incubated for 15 min at 25 °C (the same protein concentrations as the experiment of Figure 5B) in the presence of a two-fold molar excess of GroESL (lanes 6–10) or GST as control, (lanes 1–5) with respect to the highest HrcA concentration, before DNase I digestion. On the left of the autoradiograph, the numbers refer to the positions with respect to the transcriptional start site (indicated by a bent arrow). Grey boxes on the right of the autoradiographic film represent the regions of DNase I protection; black arrowheads indicate bands of hypersensitivity to DNase I digestion and numbers indicate the positions delimiting the HrcA binding site. Grey triangles indicate increasing concentrations of HrcA; 0, no HrcA protein added; dark grey box, a constant amount of GroE.

**Figure 7 biomolecules-11-01413-f007:**
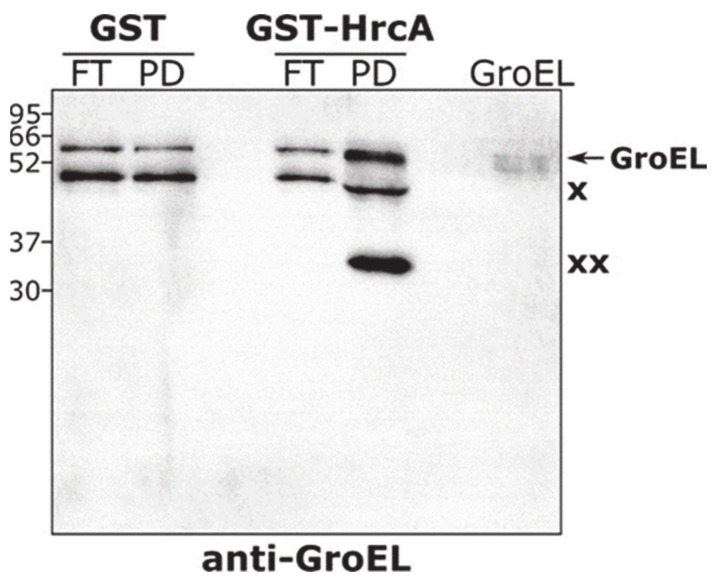
GST-pulldown assay carried out with *C. jejuni* total protein extracts. Immunoblot analysis of samples obtained from the GST-pulldown assay performed with *C. jejuni* total protein extract, stained with an anti-GroEL antibody. Samples recovered from the column containing either GST alone or GST-HrcA-Glutathione-agarose slurry (lanes labeled “PD”) were separated by SDS-PAGE and transferred on a PVDF membrane, together with samples that passed through the column, without interacting with baits (i.e., flowthrough samples, labeled “FT”). In addition, an aliquot of the purified GroEL protein was included as positive control (lane labeled “GroEL”). The intensities of the GroEL-specific bands in the GST- and GST-HrcA-PD samples were quantified using the ImageQuant software: the GroEL band in the GST-HrcA PD sample is 6.2-fold more intense than the same band in the GST PD control sample. The positions of the molecular mass standards are shown on the left. Symbols: : × and ×× indicate cross-reactive bands.

**Figure 8 biomolecules-11-01413-f008:**
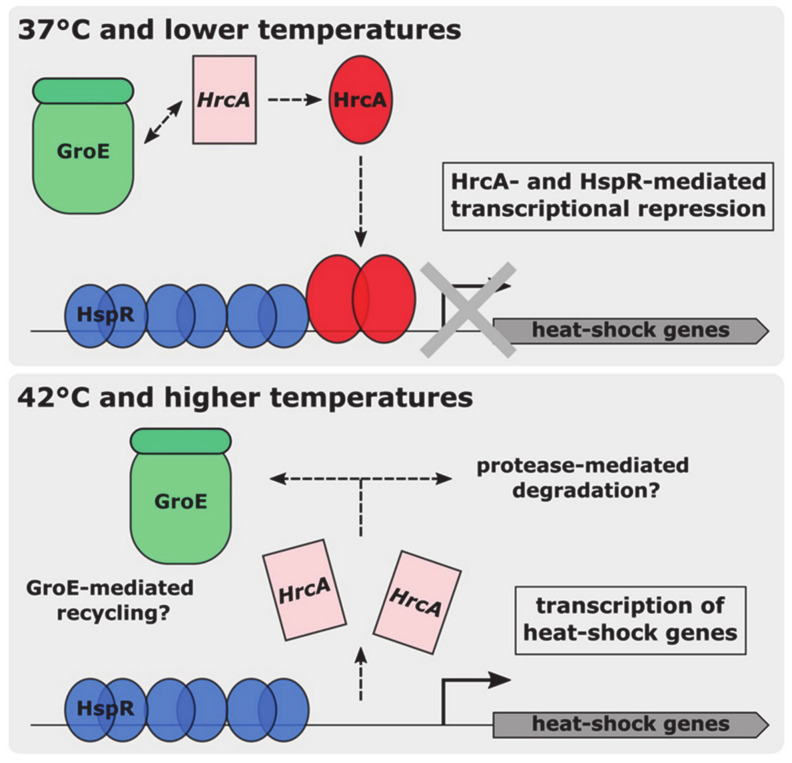
Model for heat-sensing in *C. jejuni*, representing a HrcA-HspR co-repressed promoter. Blue and red ovals represent HspR and HrcA, respectively, in their DNA-binding competent conformation, while light pink rectangles depict HrcA inactive conformation. Following an interaction with the GroE chaperonin, HrcA acquires its active conformation and binds to its operators, contributing to the transcriptional repression of heat-shock genes (upper panel). Upon a temperature increase, HrcA loses its binding capabilities and detaches from its binding sites (lower panel). It can be hypothesized that the heat-inactivated HrcA is degraded by cellular proteases. Moreover, it cannot be excluded that a fraction of inactive HrcA is actively refolded by the GroE chaperonin.

## Data Availability

Data is contained within the article or Appendix A.

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
