# Peer review of "Feeling the Heat: The *Campylobacter jejuni* HrcA Transcriptional Repressor Is an Intrinsic Protein Thermosensor"

_biomolecules, 2021, doi:10.3390/biom11101413_

Round 1
Reviewer 1 Report
The manuscript of Vesace et al. described interesting results on the regulation of heat-shock response in Campylobacter jejuni. Two regulatory proteins (HrcA and HspR) that inhibit heat-shock response were purified and assayed for DNA-binding properties to heat-shock operon sequences under different temperatures (25, 37 and 42⁰C). The results indicated that the DNA-binding property of HrcA is affected by temperature upshift. This effect is irreversible at least in vitro. On the other hand, DNA-binding of HspR to operon sequences was not affected under the same conditions. The data obtained indicates a model in which HrcA acts as an intrinsic protein thermometer sensing temperature upshifts and releasing gene expression from heat-shock operons. The data also indicated that the GroEL/GroES complex positively affects the DNA-binding property of HrcA. The work is well done and presented robust results contributing to the elucidation of a key process involved in C. jejuni regulatory response to heat-shock stress. The text is fluid, the results are appropriately described and discussed. Therefore, in my opinion, considering the quality of the data presented, the manuscript is suitable for publication. The authors could evaluate the possibility to include a figure at the end of the discussion section revisiting the model of the heat-shock response regulation by HrcA/HspR illustrating the new data described by this work.
Author Response
Point-by-point reply (in blue) to Reviewer #1
The manuscript of Versace et al. described interesting results on the regulation of heat-shock response in Campylobacter jejuni. Two regulatory proteins (HrcA and HspR) that inhibit heat-shock response were purified and assayed for DNA-binding properties to heat-shock operon sequences under different temperatures (25, 37 and 42⁰C). The results indicated that the DNA-binding property of HrcA is affected by temperature upshift. This effect is irreversible at least in vitro. On the other hand, DNA-binding of HspR to operon sequences was not affected under the same conditions. The data obtained indicates a model in which HrcA acts as an intrinsic protein thermometer sensing temperature upshifts and releasing gene expression from heat-shock operons. The data also indicated that the GroEL/GroES complex positively affects the DNA-binding property of HrcA. The work is well done and presented robust results contributing to the elucidation of a key process involved in C. jejuni regulatory response to heat-shock stress. The text is fluid, the results are appropriately described and discussed. Therefore, in my opinion, considering the quality of the data presented, the manuscript is suitable for publication.
The authors could evaluate the possibility to include a figure at the end of the discussion section revisiting the model of the heat-shock response regulation by HrcA/HspR illustrating the new data described by this work.
RESPONSE: We thank the reviewer for the positive comments and for the appreciation of the submitted manuscript. As suggested, we included a final figure (Figure 8), in which we represent a model for heat-perception in C. jejuni.
Reviewer 2 Report
The manuscript by Versace et al. provides new interesting mechanistic insights in the heat-shock response regulation in C. jejuni. In particular, the manuscript focuses on the temperature sensitivity of the two transcriptional regulators, HrcA and HspR. While these two repressors coordinate the transcriptional change in response to temperature, they show a different sensitivity to high temperature in vitro. In fact, while HspR DNA-binding activity is independent of temperature, HrcA loses the ability to bind its DNA targets upon a brief exposure to 42C. Moreover, the authors show that the DNA-binding activity of HrcA is enhanced by direct interaction with the GroE chaperonin at the permissive temperature (25C).
Overall, the experiments are well done and the manuscript is clearly written. I have relatively minor comments and questions.
1. Do the authors have any insight on the stability of HrcA at the different temperatures? Considering that the HrcA DNA-binding activity is almost irreversibly lost at 42C, might this indicate that HrcA undergoes massive degradation rather than a more moderate conformational change?
2. In figure 7, what is the ~35kDa band in the GST-HrcA PD lane? Also at line 494, the authors state that the 60kDa band (GroEL) “was much less recovered in the control sample”. It would be helpful to provide some sort of quantification to support this statement.
3. To facilitate comparison across publications, it would be helpful to introduce numbers indicating the position of the protected regions in the footprinting images.
Author Response
Point-by-point reply (in blue) to Reviewer #2
The manuscript by Versace et al. provides new interesting mechanistic insights in the heat-shock response regulation in C. jejuni. In particular, the manuscript focuses on the temperature sensitivity of the two transcriptional regulators, HrcA and HspR. While these two repressors coordinate the transcriptional change in response to temperature, they show a different sensitivity to high temperature in vitro. In fact, while HspR DNA-binding activity is independent of temperature, HrcA loses the ability to bind its DNA targets upon a brief exposure to 42C. Moreover, the authors show that the DNA-binding activity of HrcA is enhanced by direct interaction with the GroE chaperonin at the permissive temperature (25°C).
Overall, the experiments are well done and the manuscript is clearly written. I have relatively minor comments and questions.
- Do the authors have any insight on the stability of HrcA at the different temperatures? Considering that the HrcA DNA-binding activity is almost irreversibly lost at 42°C, might this indicate that HrcA undergoes massive degradation rather than a more moderate conformational change?
RESPONSE: We do not have any experimental data on the stability of HrcA protein at different temperatures. We have just empirically observed that C. jejuni HrcA shows a strong tendency to form inclusion bodies when overexpressed in a heterologous host like E. coli and is prone to form aggregates in solution, especially at ambient temperature or above (Palombo et al, Microorganisms, 2020). We reported these observations in the manuscript, at the beginning of paragraph 3.4 (lines 494-497). The possibility that the protein goes through massive degradation rather than a major conformational change (or the combination of both processes for CtsR of low-GC Gram-positive bacteria as reported by Elsholz et al, EMBO J, 2010) is interesting, but needs further experimental efforts to explore this hypothesis. To further highlight the relevance of this hypothesis, we have included the possible temperature-induced degradation of HrcA by means of cellular proteases in the model represented in the new Figure 8.
- In figure 7, what is the ~35kDa band in the GST-HrcA PD lane? Also at line 494, the authors state that the 60-kDa band (GroEL) “was much less recovered in the control sample”. It would be helpful to provide some sort of quantification to support this statement.
RESPONSE: Unfortunately, we do not have any indication suggesting the identity of the cross-reactive band clearly visible in the GST-HrcA pulldown lane of Figure 7. To support our statement about the clear enrichment of the specific GroEL signal in the GST-HrcA pulldown sample compared to the control lane, we have quantified the western blot signals. It turns out that the GroEL-specific signal in the GST-HrcA pulldown lane is 6.2-fold higher that the signal obtained in the GST pulldown control sample. We reported this information in the figure legend (lines 546-551).
- To facilitate comparison across publications, it would be helpful to introduce numbers indicating the position of the protected regions in the footprinting images.
RESPONSE: We agree with this observation and accordingly we revised all the footprinting figures, introducing, for each assay, the coordinates of the promoter with respect to the transcriptional start site and the position of the protected regions.
Reviewer 3 Report
In this manuscript, the authors focus on the transcriptional regulator HrcA that acts as a heat shock repressor. They show that a heat shock to 420C impairs its DNA binding activity on target promoter, which is in contrast to that of HspR. The authors also demonstrate that the DNA-binding activity is modulated by chaperone GroE and involves a direct interaction between GroE and HrcA. The work is performed using recombinant histidine tagged HrcA, HspR GroES/L or GST-tagged HrcA or HspR, with EMSA and DNase I footprinting to show DNA binding activity, and GST pulldown assays to demonstrate protein/protein interactions between HrcA and GroEL.
It seems that all binding sites of HrcA and HspR are already known on pgro and pclpB promoters but the novelty of this work consists in looking at the effect of temperature, specifically heat shock from ambiant to 37 or 42oC. The authors seem to have conducted similar studies on related promoters, including HrcA of Helicobacter pylori and seem to be expert in their techniques.
Most conclusions are very well supported by the data but a few simple experiments for which the authors have all tools, reagents and techniques were not performed though they would enhance the manuscript and clarify if the irreversible effect of heat shock on HrcA is an in vitro artefact rather than a physiological phenomenon.
The manuscript is generally well written but could be shortened in some areas and some ambiguous areas require minor clarifications.
Specific comments: (no particular order)
- The work is interesting as it focuses on transition from ambient to 37 or 42C, which in this reviewer’s eyes, related to host colonization (avain hosts are at 42oC) rather than simple heat shock where 42oC is just a tool with no physiological relevance but this angle is not discussed. As 42oC is the avian temperature and CJ is a commensal in this context, what is referred to as Heat shock would actually be a chronic status in birds and poultry.
Actually in several instances, the authors refer to “physiological conditions” without referring to what host (and thus temperature) is considered, so it is ambiguous.
- Abstract: “is finalized to” better changed to “results in”
“different mechanism” should probably be plural.
- Some minor English revisions needed all along, starting in abstract, just to correct grammar and remove ambiguities.
- Not sure I would say in abstract that there would be an indirect mechanism of sensing temperature for HspR since there is no effect at 42oC. Rather it could be that the temperature it is sensing is not 42oC. Maybe it is rather sensing a switch to a different temperature. Later in text, it becomes apparent that several temperatures are tested. Be more specific in abstract as it reads as if only 42oC was ever tested.
- Make abstract more explicit about the “impairment” and spell out that it ends up in de-repression of GroES/L and other HSPs.This non-specialist reviewer got confused until she saw fig 1.
- Nice Fig 1 that depicts the organization and repression. Good for non specialists.
- line 77: “under physiological conditions” in not valid. In a human that 37oC. In any bird, that’s 42oC, and thus the heat shock response would kick in at all times in a bird, not being a shock but a chronic response.
- Same comment line 84: normal conditions of growth are what temperature?
- Line 88: “induction” is not correct. It is only a de-repression.
- Line 243: “was passed through a column”: what kind of column are we talking about?
- Lines 268-270: Why decide to start the incubation at 25oC and then move up? Would the results differ if all reactions were set on ice with direct incubation at their target temperature? What happens when the 42oC exposure is sustained (like in a Campy hosted in a chicken)?
- Fig 2a: The EMSA data showing HrcA binding to Pgro at 25, 37 and 42oC are very clear and show the loss of HrcA binding at 420C.
- Fig 2b is very clear too the differences between hypersensitive and protective areas (for DNase I digestion) are clear cut and the effect of temperature also clear cut. At 42oC, more extensive areas get digested (showing more bands), indicating that the protection afforded by prior HrcA binding is gone.
Isn’t this type of experiment meant to actually map which nucleotides the binding occurs upon instead of just concluding that binding changed? I assume some parallel sequencing usually gets done? Why not here. Then what do we conclude that would be extra compared with EMSA? Please specify
Also, for non specialist, for DNase footprinting, I get it that we see protection when HrcA binds to DNA. But why do we ever see regions of hypersensitivity that are HrcA dependent developing when the protein is/stays bound? Can a word to that effect be added. It is the same for HspR, so likely it is something obvious for people in the field.
- This reviewer is not a specialist in this field and does not know prior art about CJ HrcA and HspR DNA binding activity. Reference 16 already indicates that there are 3 HspR binding sites on Pgro as indicated line 335-336. So, this was known. The novelty here is to test the effect of temperature. Maybe that needs to be stated more clearly.
- EMSA and DNaseI footprinting data of high quality also for HspR too.
- Lines 353-355: hypothesis of how the dual controlled promoter would work in light of differential temperature effects on DNA binding of HrcA and HspR is clear. But is should come later, after the study of the lack of temperature dependency of HspR binding to DNA is finished (after the pClpB promoter EMSA and DNaseI work).
- Line 356 is ambiguous: the wording “behavior of HspR on exclusively repressed promoters” puts emphasis on “repressed” while I assume the authors meant promoters repressed exclusively by HspR instead of both hspR and HrcA. I think the location of “exclusively” is not right and is misleading. Maybe it would be” behavior of repressed promoters exclusively controlled by hspR”. It gets clarified later (line 357) but threw me off initially.
- It is good that the authors also tried a higher temperature for HspR. Overall, they covered 25 37 42 and 45oC. The abstract does not reflect that and implied only 37 vs 42oC. Add the info.
- Line 408-409: the 10 min recovery step seems short. Is it usually that short in this type of work in other bacteria or with other transcriptional regulators? Have the authors tried a longer recovery (say up to 30 min) and/or also assessed recovery at a higher temperature (37oC might allow more structural recovery that colder temperature)?
- Agreed with conclusion for HrcA that in absence of other cellular factors, DNA binding activity of HrcA is irreversible affected by heating in vitro. Though it needs to be put in biological context. When temperature shifts down again in vivo, newly synthesized basal level of HrcA can be functional and will come back to serve as repressor. Or there will be assisted refolding going on. Maybe a word here.
- Line 446: missing “of”
- Fig 6 clearly show enhanced DNA binding activity of HrcA in the presence of GroES/L. However Fig S2 is not convincing. The difference in signal with groE is very small and could reflect the overall amount of probe that appears higher in the GroE lanes.
- Line 473: “direct protein protein interaction” is a little misleading as it would imply the HrcA and GroEL need to be bound together while the DNA binding occurs. It is rather that groEL interacts with HrcA to help fold it and stabilize it, but I doubt it stays bound to it during the DNA binding, right? If assays have been tried with a HrcA/GroE ratio >50 or100, then that would show an “enzymatic” activity of GroEL whereby the permanent binding to HrcA is not needed. Please comment, explain. Even GST-pull down does not require a permanent interaction, does it? As long as there is enough interacting partner to saturate the GST-tagged bait, one can get the partner out, right?
- For fig 7, I’d like to see the Ponceau stain of the blot before probing, or if not possible, a matching Coomassie gel to assess total loadings and specificity of GroEL signal. It is quite important when one is talking about an “enrichment” as is the case here as opposed to an “all or nothing” situation.
Plus labeling what band is what with arrows and a text box would help since there are 3 reactive bands in total and only 1 expected. I am not seeing a comment on what is the strong band at 35 kDa?
- Was this pull-down done with HspR too to assess specificity? In mat/met, GST-tagged HspR was mentioned. Line 515 said heat detection is mediated by yet unidentified protein partner. The authors could carry out a similar pull down and assess the presence of a new partner by MS of differing bands. This would really enhance the paper.
- To address the item of cooperative binding of line 522, a round of EMSAs on both pgro and pclpB promoters with different ratios of HspR and HrcA would have been nice. The authors have all the tools available (proteins, probes, working EMSA assays). It is odd this was not done and would be a nice addition. It would test what is written as an hypothesis on lines 524-525 in the discussion. I think it should be added.
- The temperature reversal experiments should have been done in the presence of GroEL too, as in vivo, there is always groEL around. Since HrcA is intrinsically unstable, the “irreversible” nature of the heat shock may not be physiologically relevant. This could explain why HrcA is an exception compared with others (as discussed lines 531-535). I strongly suggest to perform this experiment. Again authors have all tools necessary.
- Line 540-541: how homologous/identical are the C. jejuni and H. pylori HrcA molecules. The authors also studied the HP HrcA. It was studied at 42oC too, but this is never physiologically relevant for HP in a host, while it is for CJ. Is the HP homologue as unstable as the CJ one? Did it need GroEL at all? Comments are needed.
- A final model figure showing the role of GroEL on HrcA strcutrue and activity could enhance the paper.
- Overall, since the same assays are used many times, the data description could be shortened a bit in the later sections.
Author Response
Point-by-point reply (in blue) to Reviewer #3
In this manuscript, the authors focus on the transcriptional regulator HrcA that acts as a heat shock repressor. They show that a heat shock to 42°C impairs its DNA binding activity on target promoter, which is in contrast to that of HspR. The authors also demonstrate that the DNA-binding activity is modulated by chaperone GroE and involves a direct interaction between GroE and HrcA. The work is performed using recombinant histidine tagged HrcA, HspR GroES/L or GST-tagged HrcA or HspR, with EMSA and DNase I footprinting to show DNA binding activity, and GST pulldown assays to demonstrate protein/protein interactions between HrcA and GroEL.
It seems that all binding sites of HrcA and HspR are already known on pgro and pclpB promoters but the novelty of this work consists in looking at the effect of temperature, specifically heat shock from ambient to 37 or 42°C. The authors seem to have conducted similar studies on related promoters, including HrcA of Helicobacter pylori and seem to be expert in their techniques.
Most conclusions are very well supported by the data, but a few simple experiments for which the authors have all tools, reagents and techniques were not performed though they would enhance the manuscript and clarify if the irreversible effect of heat shock on HrcA is an in vitro artefact rather than a physiological phenomenon.
The manuscript is generally well written but could be shortened in some areas and some ambiguous areas require minor clarifications.
Specific comments: (no particular order)
- The work is interesting as it focuses on transition from ambient to 37 or 42°C, which in this reviewer’s eyes, related to host colonization (avian hosts are at 42°C) rather than simple heat shock where 42°C is just a tool with no physiological relevance but this angle is not discussed. As 42°C is the avian temperature and CJ is a commensal in this context, what is referred to as Heat shock would actually be a chronic status in birds and poultry.
RESPONSE: In our opinion, this is an improper interpretation of the heat-shock response, which in bacteria is characterized by a rapid and transient transcriptional response to allow the cell to adapt "quickly" to the new condition. In bacteria, it has been documented that the heat shock response consists in the rapid increase of the transcription of specific genes (reaches a maximum transcript peak after 15-20 min from the stimulus) and then slowly decreases towards a level of steady-state just above the level present at the non-stress temperature (about 1 hour after the stimulus). In the case of C. jejuni, which can survive in food (room temperature or chilled), humans (37 ° C), and chicken (42 ° C), the heat shock response was observed, similar to other bacteria, in the transition from 37 to 42 ° C and is considered as a signal to establish avian infection. Please see lines 54-57 and 71-75 of the Introduction section.
Actually, in several instances, the authors refer to “physiological conditions” without referring to what host (and thus temperature) is considered, so it is ambiguous.
RESPONSE: We agree with the reviewer and we revised all the text accordingly, clarifying in each case this aspect.
- Abstract: “is finalized to” better changed to “results in”
RESPONSE: We agree and revised the text accordingly.
- “different mechanism” should probably be plural.
RESPONSE: Revised.
- Some minor English revisions needed all along, starting in abstract, just to correct grammar and remove ambiguities.
RESPONSE: We revised the text, correcting errors and removing ambiguities.
- Not sure I would say in abstract that there would be an indirect mechanism of sensing temperature for HspR since there is no effect at 42°C. Rather it could be that the temperature it is sensing is not 42°C. Maybe it is rather sensing a switch to a different temperature. Later in text, it becomes apparent that several temperatures are tested. Be more specific in abstract as it reads as if only 42°C was ever tested.
RESPONSE: We agree with this point and revised the abstract, in which we explicitly state that HspR appears to be unaffected by temperature increases up to 45°C (lines 20-23: “On the other hand, we demonstrate that the DNA-binding activity of HspR, which controls, in combination with HrcA, the transcription of chaperones’ genes, is unaffected by heat-stress up to 45°C, portraying this master repressor as a rather stable protein.”).
- Make abstract more explicit about the “impairment” and spell out that it ends up in de-repression of GroES/L and other HSPs. This non-specialist reviewer got confused until she saw fig 1.
RESPONSE: We revised the abstract, making more explicit the role of HrcA and HspR as repressors of heat-shock genes (lines 14-20: “In Campylobacter jejuni the response to heat-shock is transcriptionally controlled by a regulatory circuit involving two repressors, HspR and HrcA. In the present work we show that the heat-shock repressor HrcA acts as an intrinsic protein thermometer. We report that a temperature upshift up to 42°C negatively affects HrcA DNA-binding activity to a target promoter, a condition required for de-repression of regulated genes. Furthermore, we show that this impairment of HrcA binding at 42°C is irreversible in vitro, as DNA-binding was still not restored by reversing the incubation temperature to 37°C.”).
- Nice Fig 1 that depicts the organization and repression. Good for non-specialists.
- line 77: “under physiological conditions” is not valid. In a human that 37°C. In any bird, that’s 42°C, and thus the heat shock response would kick in at all times in a bird, not being a shock but a chronic response.
RESPONSE: The model of heat-shock genes’ regulation described in Figure 1 is based on previous articles describing HrcA and HspR interactions with target promoters (Palombo et al, Microorganisms, 2020) and transcriptional studies (Stintzi et al, J. Bacteriol. 2003). In this latter work, the authors compared the transcriptome of C. jejuni cells exposed to 42°C, compared to control cells grown at 37°C. Following the reviewer’s suggestion, we clarify this point (lines 86-88: “At temperatures up to 37°C, HspR binds and represses its own promoter (Pcbp), as well as the regulatory region controlling the transcription of the clpB containing operon (Pclp).”).
- Same comment line 84: normal conditions of growth are what temperature?
RESPONSE: We revised this sentence as follows (lines 93-96): “This negative regulatory circuit implies, at 37°C which represent the normal temperature for the human host, a transcriptional repression of HSP-encoding genes operated by HspR and HrcA repressors bound to their operators within the heat-shock promoters.”.
- Line 88: “induction” is not correct. It is only a de-repression.
RESPONSE: This was indeed a mistake; it has been corrected.
- Line 243: “was passed through a column”: what kind of column are we talking about?
RESPONSE: In this step, we used empty gravity flow columns which retain the chromatographic resin (glutathione-agarose slurry bound to GST, in this case). We revised the text, clarifying this point (lines 262-264: ”After that, the sample was passed through an empty column which retained the resin, and the cleared extract was collected for the pulldown experiment.”).
- Lines 268-270: Why decide to start the incubation at 25°C and then move up? Would the results differ if all reactions were set on ice with direct incubation at their target temperature? What happens when the 42°C exposure is sustained (like in a Campy hosted in a chicken)?
RESPONSE: Indeed, we do not have any experimental observation to hypothesize the behavior of the proteins if incubated at their target temperature directly after setting up the reactions on ice. We decided to start the incubation for 10 min at 25°C because we have already demonstrated that, in these conditions, HrcA and HspR bind to their specific operators on the selected promoter region (Palombo et al, Microorganisms, 2020). Indeed, in this work we wanted to study the effect of temperature on HrcA and HspR DNA-binding capabilities and, for this purpose, we decided to promote binding of proteins to the DNA and observe if their interaction is affected by moving the reaction to 37°C (normal temperature of the human host) or 42°C (normal avian temperature). We made more explicit this point by modifying the text as follows (lines 286-291): “In order to investigate if temperature is able to influence HrcA activity, DNA-binding of the protein was promoted by incubating the reactions at 25°C for 10 min (a condition previously used to study HrcA and HspR interaction with the DNA in [16]), then samples were shifted at 25°C, 37°C or 42°C for an additional period of time of 10 min, before separation of the complexes through native polyacrylamide gel electrophoresis.”.
In this work, we did not extend the exposure at 42°C beyond the 10 min time reported in the manuscript. We do not know what happens in vivo upon sustained exposure at 42°C, as for example in an avian host. One possibility is that the increased amount of repressor molecules that accumulates inside the cell (above a repression threshold) is sufficient to re-establish the transcriptional repression of heat-shock genes. It has been observed, in fact, that once the organism has been exposed to a higher temperature, it adapts to the new temperature, and the amount of heat-shock proteins decreases to a steady-state level that is frequently greater than the initial basal level (Roncarati et al, FEMS Microbiology Reviews, 2017). We do not know if this is the case also for C. jejuni shifted to 42°C for an extended period of time.
- Fig 2a: The EMSA data showing HrcA binding to Pgro at 25, 37 and 42°C are very clear and show the loss of HrcA binding at 42°C.
- Fig 2b is very clear too, the differences between hypersensitive and protective areas (for DNase I digestion) are clear cut and the effect of temperature also clear cut. At 42°C, more extensive areas get digested (showing more bands), indicating that the protection afforded by prior HrcA binding is gone.
Isn’t this type of experiment meant to actually map which nucleotides the binding occurs upon instead of just concluding that binding changed? I assume some parallel sequencing usually gets done? Why not here. Then what do we conclude that would be extra compared with EMSA? Please specify
RESPONSE: Both DNase I footprinting and EMSA assays are molecular biology techniques widely used to characterize protein-DNA interactions. In this work, we decided to use both techniques to study the effect of temperature on HrcA and HspR DNA-binding activity in order to get information from two independent methods. Moreover, DNAse I footprint experiment is used to map the region of protein interaction and whether the DNA undergoes possible conformational changes (hypertensive sites), whereas EMSA experiment is informative only of the binding.
Footprinting provides information on the precise position of the binding site of a DNA-binding protein of interest. To this aim, a G+A (Maxam & Gilbert) sequencing reaction is usually run in parallel to the DNase I footprinting reactions. To include the information about the position of protein binding sites, we revised all the footprinting figures, introducing for each assay the coordinates of the promoter with respect to the transcriptional start site and the position of the protected regions.
- Also, for non-specialist, for DNase footprinting, I get it that we see protection when HrcA binds to DNA. But why do we ever see regions of hypersensitivity that are HrcA dependent developing when the protein is/stays bound? Can a word to that effect be added? It is the same for HspR, so likely it is something obvious for people in the field.
RESPONSE: To clarify this point, we revised the text describing the results obtained in the first Footprinting presented in this work (lines 322-327): “As shown in Figure 2b, upon the addition of increasing concentrations of protein to the reaction at 25°C, an area of protection indicative of HrcA binding to the Pgro probe (marked by boxes) appeared (Figure 2b, lanes 1- 4). In the same experiment, diverse DNase I hypersensitive sites were detected (marked by arrowheads). Likely, these arise from a conformational change of the probe following protein-DNA interaction.”.
- This reviewer is not a specialist in this field and does not know prior art about CJ HrcA and HspR DNA binding activity. Reference 16 already indicates that there are 3 HspR binding sites on Pgro as indicated line 335-336. So, this was known. The novelty here is to test the effect of temperature. Maybe that needs to be stated more clearly.
RESPONSE: A sentence clarifying this information has been added in the Introduction section (lines 130-133): “Recently, we determined the C. jejuni HrcA and HspR interactions on heat shock promoters by high-resolution DNase I footprints, showing that while DNA-binding of HrcA covers a compact region, HspR interacts with multiple high- and low-affinity binding sites [16].”.
- EMSA and DNaseI footprinting data of high quality also for HspR too.
- Lines 353-355: hypothesis of how the dual controlled promoter would work in light of differential temperature effects on DNA binding of HrcA and HspR is clear. But it should come later, after the study of the lack of temperature dependency of HspR binding to DNA is finished (after the pClpB promoter EMSA and DNaseI work).
RESPONSE: As suggested by the reviewer, we revised the second part of paragraph 3.2, introducing the hypothesis about the way in which HrcA and HspR co-regulated promoters perceive the temperature increase in the final section (lines 441-453). Moreover, we have carried out a new EMSA experiment (Figure S3), in which we included a reaction in which HrcA and HspR were simultaneously incubated with the Pgro probe and monitored the DNA-binding behavior of both repressors at different temperatures.
- Line 356 is ambiguous: the wording “behavior of HspR on exclusively repressed promoters” puts emphasis on “repressed” while I assume the authors meant promoters repressed exclusively by HspR instead of both hspR and HrcA. I think the location of “exclusively” is not right and is misleading. Maybe it would be” behavior of repressed promoters exclusively controlled by hspR”. It gets clarified later (line 357) but threw me off initially.
RESPONSE: We agree that this sentence could be misleading and we revised this part as follows (lines 378-379): “…we wanted to assess the temperature-dependent DNA-binding behavior of HspR on promoters exclusively controlled by this master repressor.”.
- It is good that the authors also tried a higher temperature for HspR. Overall, they covered 25 37 42 and 45oC. The abstract does not reflect that and implied only 37 vs 42oC. Add the info.
RESPONSE: We agree with this reviewer’s comment and we revised this part of the abstract accordingly (lines 20-23).
- Line 408-409: the 10 min recovery step seems short. Is it usually that short in this type of work in other bacteria or with other transcriptional regulators? Have the authors tried a longer recovery (say up to 30 min) and/or also assessed recovery at a higher temperature (37°C might allow more structural recovery that colder temperature)?
RESPONSE: According to Servant et al. (Proc. Natl. Acad. Sci. USA, 2000), the 10 min recovery step can be considered a long time, as for the RheA transcriptional regulator of Streptomyces albus has been used a recovery time of 1 min. Also, in other examples found in literature (Hurme at al, Cell, 1997; Herbst, PLoS Pathog, 2009) a similar experimental layout was adopted, employing a few-minutes long recovery time after heat-challenge. This reaction is believed to be very fast. The above reference concerning RheA characterization has been added on line 481.
- Agreed with conclusion for HrcA that in absence of other cellular factors, DNA binding activity of HrcA is irreversible affected by heating in vitro. Though it needs to be put in biological context. When temperature shifts down again in vivo, newly synthesized basal level of HrcA can be functional and will come back to serve as repressor. Or there will be assisted refolding going on. Maybe a word here.
RESPONSE: We agree with the reviewer’s suggestion and we extended the end of the discussion section accordingly (lines 645-647: “When temperature shifts down again, this hypothetical fraction of GroE-refolded HrcA protein would contribute, together with the newly synthesized polypeptide, to restore the transcriptional repression of heat-shock genes.”).
- Line 446: missing “of”
RESPONSE: Corrected.
- Fig 6 clearly show enhanced DNA binding activity of HrcA in the presence of GroES/L. However, Fig S2 is not convincing. The difference in signal with GroE is very small and could reflect the overall amount of probe that appears higher in the GroE lanes.
RESPONSE: This is likely due to a higher sensitivity of the DNase I footprints than EMSA. With the same amount of HrcA in Fig. S2, the band intensity in lane 9 is more intense than that in lane 4, in agreement with results in Fig. 6 and supporting the hypothesis that HrcA binding is stimulated by the presence of GroE in the reaction.
- Line 473: “direct protein-protein interaction” is a little misleading as it would imply the HrcA and GroEL need to be bound together while the DNA binding occurs. It is rather that groEL interacts with HrcA to help fold it and stabilize it, but I doubt it stays bound to it during the DNA binding, right? If assays have been tried with a HrcA/GroE ratio >50 or100, then that would show an “enzymatic” activity of GroEL whereby the permanent binding to HrcA is not needed. Please comment, explain. Even GST-pull down does not require a permanent interaction, does it? As long as there is enough interacting partner to saturate the GST-tagged bait, one can get the partner out, right?
RESPONSE: We agree with all the comments raised by the reviewer about HrcA-GroE interaction. We share with the reviewer the idea that GroE transiently interacts with HrcA and promotes its folding to enhance HrcA DNA-binding affinity. Accordingly, we revised the text as follows (lines 527-530): “In conclusion, these experiments suggest that GroE is able to enhance HrcA DNA binding affinity for its binding site on Pgro, thereby contributing to the transcriptional repression of the HrcA controlled promoters.”. In addition, in the final model represented in Figure 8, we propose the mechanism of the transient HrcA-GroE interaction followed by the interaction of the properly folded HrcA (alone, not as a complex with GroE) with its binding site on a target promoter.
- For fig 7, I’d like to see the Ponceau stain of the blot before probing, or if not possible, a matching Coomassie gel to assess total loadings and specificity of GroEL signal. It is quite important when one is talking about an “enrichment” as is the case here as opposed to an “all or nothing” situation.
RESPONSE: Since the amount of proteins recovered from a pull-down assay is very limited, a Ponceau staining of the blot (as well as a Coomassie staining of a matching gel) would not give any signal (except for the GST and GST-HrcA bait proteins, used in equimolar amounts). For this reason, we consider the non-specific cross-reactive band visible in all the samples (labelled X in Figure 7) a good loading control. To support our statement about the clear enrichment of the specific GroEL signal in the GST-HrcA pulldown sample compared to the control lane, we have quantified the western blot signals, using the above-mentioned cross-reactive band (X) as a normalizer. It turns out that the GroEL-specific signal in the GST-HrcA pulldown lane is 6.2-fold higher that the signal obtained in the GST pulldown control sample. We reported this information in the figure legend (lines 546-551).
- Plus, labeling what band is what with arrows and a text box would help since there are 3 reactive bands in total and only 1 expected. I am not seeing a comment on what is the strong band at 35 kDa?
RESPONSE: We revised Figure 7, including an arrow and two crosses which highlight the specific GroEL signal and non-specific cross-reactive bands, respectively. Concerning the 35 kDa cross-reactive band clearly visible in the GST-HrcA pulldown lane, we do not have any indication suggesting its identity.
- Was this pull-down done with HspR too to assess specificity? In mat/met, GST-tagged HspR was mentioned. Line 515 said heat detection is mediated by yet unidentified protein partner. The authors could carry out a similar pull down and assess the presence of a new partner by MS of differing bands. This would really enhance the paper.
RESPONSE: We did not carry out pull-down assay with GST-HspR, indeed the reference to this fusion protein in the paragraph 2.6 is a mistake. We corrected the text accordingly.
We agree with the reviewer that an approach consisting of a pull-down experiment followed by MS analysis would be highly informative towards the identification of possible new partner(s). However, this is a new line of investigation that requires a method not available in our laboratory. We plan to tackle it in a future study, when valuable collaborations will be established.
- To address the item of cooperative binding of line 522, a round of EMSAs on both Pgro and PclpB promoters with different ratios of HspR and HrcA would have been nice. The authors have all the tools available (proteins, probes, working EMSA assays). It is odd this was not done and would be a nice addition. It would test what is written as an hypothesis on lines 524-525 in the discussion. I think it should be added.
RESPONSE: We agree that competitive DNA-binding experiments with HrcA and HspR on co-regulated Pgro promoter would offer new insights on the reciprocal effect of the regulators upon temperature variations. To address this issue in the time frame allowed, we set up an EMSA assay in which we monitored the DNA-binding behavior of HrcA and HspR at different temperature, including reactions in which we simultaneously added both proteins to the reaction. Results have been presented at the end of paragraph 3.2 (lines 444-453), shown in Figure S3 and discussed in the Discussion section (lines 584-589).
- The temperature reversal experiments should have been done in the presence of GroEL too, as in vivo, there is always GroEL around. Since HrcA is intrinsically unstable, the “irreversible” nature of the heat shock may not be physiologically relevant. This could explain why HrcA is an exception compared with others (as discussed lines 531-535). I strongly suggest to perform this experiment. Again, authors have all tools necessary.
RESPONSE: As suggested by the reviewer, we performed the temperature reversal experiment in the presence of the GroE chaperonin complex and monitored its effect on DNA-binding capacity of heat-shocked HrcA by means of DNase I footprinting assay. The result obtained has been included in Figure S4 and discussed in the Discussion section (lines 633-640: “Considering the HrcA heat-shock induced loss of DNA binding activity presented here, a possibility is that in GroE can assist also HrcA refolding, lost upon a heat-challenge, when permissive conditions are restored (Figure 8). To test this hypothesis, we assayed in vitro the ability of GroE to restore the DNA-binding activity of HrcA, lost upon exposure to 42°C. The footprinting assay shown in Figure S4 suggests that, in the experimental conditions used, the GroE chaperonin has a barely visible effect on HrcA functionality. However, we cannot rule out the hypothesis that in vivo, GroE takes part in the regulation of HrcA DNA-binding capability also upon a heat-challenge.”).
- Line 540-541: how homologous/identical are the C. jejuni and H. pylori HrcA molecules. The authors also studied the HP HrcA. It was studied at 42°C too, but this is never physiologically relevant for HP in a host, while it is for CJ. Is the HP homologue as unstable as the CJ one? Did it need GroEL at all? Comments are needed.
RESPONSE: We revised the text accordingly (lines 613-618): “This common behavior of the two HrcA proteins from H. pylori and C. jejuni is not surprising, as they share 27.3% of amino acids identity, are both unstable proteins, display a strong tendency to form aggregates in vitro when incubated at temperatures above 40°C. Furthermore, the DNA-binding activity of H. pylori HrcA is stimulated by the GroE chaperonin under normal conditions of growth, as well as following a heat-challenge to 42°C [30,38].”.
- A final model figure showing the role of GroEL on HrcA structure and activity could enhance the paper.
RESPONSE: As suggested, we included a final figure (Figure 8), in which we represent a model for heat-perception in C. jejuni, showing the effect of the GroE chaperonin on HrcA structure and activity.
- Overall, since the same assays are used many times, the data description could be shortened a bit in the later sections.
RESPONSE: We agree and attempted revisions, accordingly.
On line 382 we deleted “following a 10-minutes incubation at 25°C”.
On line 441 we deleted “used above to assess DNA-binding capabilities of both HrcA and HspR, is co-regulated by the two repressors and that”.
Round 2
Reviewer 3 Report
The authors added the experiments I requested, with the figures, plus added the model figure and added all explanations required to clarify details and make the writing more accessible to non specialists. Other DNA footprinting figures were also enhanced by promoter position info. This helps a lot. Great job, thank you. Paper ready to go as is.